# Automated Collection and Analysis of Infrared Thermograms for Measuring Eye and Cheek Temperatures in Calves

**DOI:** 10.3390/ani10020292

**Published:** 2020-02-12

**Authors:** Gemma Lowe, Brendan McCane, Mhairi Sutherland, Joe Waas, Allan Schaefer, Neil Cox, Mairi Stewart

**Affiliations:** 1InterAg, Ruakura Research Centre, Hamilton 3214, New Zealand; stewartmairi42@gmail.com; 2School of Science, The University of Waikato, Hamilton 3216, New Zealand; joe@waas.ca; 3Department of Computer Science, University of Otago, Dunedin 9016, New Zealand; mccane@cs.otago.ac.nz; 4AgResearch Ltd., Ruakura Research Centre, Hamilton 3214, New Zealand; Mhairi.Sutherland@agresearch.co.nz; 5Animal Inframetrics, Box 5451, Lacombe, AB T4L 1X2, Canada; alschaef@telus.net; 6NeilStat Ltd., 9 Ngaere Ave, Hamilton 3210, New Zealand; neil.cox@xtra.co.nz

**Keywords:** Infrared thermography, algorithm validation, automated systems, early disease detection, calves

## Abstract

**Simple Summary:**

In response to an increasing shift towards automation in the livestock industry, it is important that, as automated systems are developed, they include the capability for automated monitoring of animal health and welfare. As an alternative to current manual methods, this study reports on the development and validation of an algorithm for automated detection and analysis of the eye and cheek regions from thermal infrared images collected automatically from calves. Such algorithms are essential for the integration of infrared thermography (IRT) technology into automated systems to noninvasively monitor calf health and welfare.

**Abstract:**

As the reliance upon automated systems in the livestock industry increases, technologies need to be developed which can be incorporated into these systems to monitor animal health and welfare. Infrared thermography (IRT) is one such technology that has been used for monitoring animal health and welfare and, through automation, has the potential to be integrated into automated systems on-farm. This study reports on an automated system for collecting thermal infrared images of calves and on the development and validation of an algorithm capable of automatically detecting and analysing the eye and cheek regions from those images. Thermal infrared images were collected using an infrared camera integrated into an automated calf feeder. Images were analysed automatically using an algorithm developed to determine the maximum eye and cheek (3 × 3-pixel and 9 × 9-pixel areas) temperatures in a given image. Additionally, the algorithm determined the maximum temperature of the entire image (image maximum temperature). In order to validate the algorithm, a subset of 350 images analysed using the algorithm were also analysed manually. Images analysed using the algorithm were highly correlated with manually analysed images for maximum image (R^2^ = 1.00), eye (R^2^ = 0.99), cheek 3 × 3-pixel (R^2^ = 0.85) and cheek 9 × 9-pixel (R^2^ = 0.90) temperatures. These findings demonstrate the algorithm to be a suitable method of analysing the eye and cheek regions from thermal infrared images. Validated as a suitable method for automatically detecting and analysing the eye and cheek regions from thermal infrared images, the integration of IRT into automated on-farm systems has the potential to be implemented as an automated method of monitoring calf health and welfare.

## 1. Introduction

Over time, the livestock industry has seen a significant change in response to an increasing reliance on automated systems. This reliance has largely been driven by a need to reduce labour costs and has resulted in the development of automated systems such as the robotic milking and automated calf feeder systems seen in modern dairy farming systems [1]. This increasing level of automation in the livestock industry [2] has resulted in a less “hands-on” approach to farming, and additionally, there are fewer experienced stock people in the industry [3]. These effects of automation, along with increasing herd sizes and less individual animal contact, can result in a reduced ability to monitor and identify animals displaying signs of compromised health and welfare. It is therefore important as part of the future development of automated systems that they are designed to incorporate methods which provide the capability to reliably monitor animal health and welfare on-farm.

Infrared thermography (IRT) is a technology which has the potential to be integrated into automated systems for monitoring animal health and welfare [3]. IRT detects the amount of infrared energy an object radiates; the more infrared energy, the greater the temperature of the object [4]. IRT is a noninvasive, remote method of measuring an animal’s surface temperature [5], where the temperatures detected and the distributions may be associated with underlying physiological, metabolic and behavioural processes and mechanisms [6]. There is the potential that, by measuring changes in surface temperature, IRT can be applied as a method for detecting disease based on the detection of fever/inflammation [7]. The use of IRT in livestock and veterinary applications was most recently reviewed by Luzi et al. [8]. Applications for which IRT has been applied include the detection of bluetongue virus in sheep [9], of foot and mouth disease in mule deer [10], of rabies in raccoons [11], of thoracolumbar vertebral disk disease in dogs [12], of pregnancy in zebras and black rhinoceros [13] and of impaired meat quality in pigs [14]. With specific relevance to cattle welfare, some of the applications that IRT has been used for include diagnosing mastitis [15] and lameness [16,17] by detecting areas of inflammation and as a method for measuring stress and pain responses to procedures such as disbudding [18], castration [19], handling [20] and transport [21]. IRT has also been used in previous studies as a method for the early detection of diseases such as bovine viral diarrhea (BVD), bovine respiratory disease (BRD) [22,23,24] and neonatal calf diarrhea (NCD) [1,25] and, additionally, as a method for detecting differences in feed efficiency [26,27,28]. Previous studies have collected infrared images from various anatomical regions. Whilst investigating IRT as a method for detecting BVD for example, Schaefer et al. [22] found changes in eye temperature as early as 1 d postinfection; however, changes in nose, ear, hoof, lateral and dorsal temperatures were not significant until 5–6 days postinfection. The increased sensitivity of the eye was considered to be attributed to the blood flow being closer to the surface and thus providing a more accurate reflection of core body temperature [5].

Generally, thermal infrared images are collected and analysed manually, where an observer collects images using a handheld infrared camera (e.g., ThermaCAM S60; FLIR systems AB, Danderyd, Sweden) [1,26]. These images are then analysed using specific analysis software (e.g., ThermaCAM Researcher; FLIR systems AB, Danderyd, Sweden). This software requires observers to select the region of interest (ROI) in each image to obtain the minimum, median and maximum temperatures of the area. These manual methods are often time-consuming, especially when dealing with large groups of animals and large data sets and requires the skill of trained observers during image collection and analysis [29]. Additionally, handheld infrared cameras are often cumbersome and impractical to use and can, in some situations, cause disturbances for the animals, which can affect the results [29]. The development of automated methods for the collection and analysis of thermal infrared images would provide an alternative to current manual methods [29]. Furthermore, the development of algorithms is necessary for the successful integration of IRT into automated systems where it could then be utilized for noninvasive monitoring of animal health and welfare. The integration of IRT into automated systems could potentially enable diseased animals to be identified sooner than is currently possible based on overt clinical signs [1]. This would enable sick animals to be identified and isolated from their pen mates to prevent the spread of disease and would enable treatments to be administered sooner [1]. A system providing the ability to detect early disease onset would also facilitate decision-making abilities for farmers, reducing costs on-farm and for the industry as a whole through reduced labour, mortalities and veterinary costs, and, overall, would lead to improvements in both production and animal welfare [1].

This study was part of a larger project investigating a wide range of automated methods for early disease detection in calves. The purpose of the current project was 1) to develop an algorithm capable of detecting and analysing temperatures of the eye and cheek regions from thermal infrared images (collected automatically while calves visited an automated calf feeder) and 2) to validate the algorithm through comparison with manual methods of analysis. 

## 2. Materials and Methods 

All procedures involving animals in this study were approved by the University of Waikato Animal Ethics Committee (Protocols #955 and 985) under the New Zealand Animal Welfare Act 1999.

### 2.1. Development of Eye and Cheek Algorithm

An algorithm was developed with the capability of automatically detecting and analysing the eye and cheek from thermal infrared images collected from calves. Within each captured image, individual pixels acted as floating-point numbers, indicating temperature in degrees Kelvin. Images were converted from floating-point raw data into 8-bit grey-scale portable network graphics (PNG) images using the following conversion formula:(1)$$x^{\prime }=255\frac{\left(x-260\right)}{315-260}$$
where *x* is the original temperature, $x^{\prime }$ is the scaled pixel value, 260 is the minimum temperature and 315 is the maximum temperature. All temperatures which fell outside the minimum and maximum temperatures were clamped at those values to maximise the temperature resolution when converting the raw data into a PNG image. The original temperature (*x*) was calculated from the PNG images using the following formula:(2)$$x=\frac{x^{\prime }\left(315-260\right)}{255}+260$$

The calculation of *x* was subject to a conversion accuracy of approximately ±0.2 °C as a result of the level of error in converting from float to byte values.

The eye detection component of the algorithm used a cascade of AdaBoost classifiers using decision stumps and Haar wavelets [30] and was generated using the OpenCV (version 3.4.3) [31] utility opencv_trainscascade with the following parameters: stageType = BOOST; featureType = HAAR; width = 24; height = 24; boostType = GAB; minHitRate = 0.99; maxFalseAlarm = 0.4; maxDepth = 1; maxWeakCount = 100; featSize = 1; and mode = BASIC.

For further development of the eye detection component, the OpenCV utility required a set of training images. Training images were collected during a previous study by our group [32] using an infrared camera (Thermovision A300 (accuracy: ±2.0 °C; sensitivity: <0.05 °C; resolution: 320 × 240; temperature range: −20 °C to 120 °C; spectral range: 7.5–13 µm); FLIR Systems AB, Danderyd, Sweden) set up at an automated calf feeder (RFID Calf Feeder, A&D Reid, Temuka, New Zealand (as used previously by Lowe et al. [1])) in the same manner as the current study (as described below). Training images were collected from 23 animals, with the number of images captured for each individual varying from 162 to 252 images, yielding a total of 5250 images. For training purposes, images were split into both positive and negative examples. 

Positive examples were images that, with the exception of including some surrounding area, contained only the ROI. Including some surrounding area to the ROI has been found to improve the robustness of the detector in face detection applications [33] and hence was included in the positive training images. From the eye, maximum temperature has been found to be the most relevant diagnostic [24]; therefore, for training purposes, the location of the maximum temperature was identified manually in 1364 images. To create the positive training images, a 72 × 72-pixel sub-image was extracted from the original image based on the thermal maximum with a top-left position of (*x* − 12, *y* − 36) and a bottom right position of (*x* + 60, *y* + 36). Once extracted, these sub-images were resized for efficiency purposes to 24 × 24 pixels for the training process. The process of creating the positive training sub-images is shown in Figure 1.

Negative training examples did not include the whole eye region but consisted of parts of an eye. In addition to images which only partially included the eye, negative training examples included those in which the eye was fully or partially closed, as these types of images reduce the ability to acquire an accurate temperature measurement. Examples of negative training images are shown in Figure 2.

The training process resulted in an 8-stage cascade, with each stage consisting of 3, 3, 3, 5, 4, 6, 8 and 9 weak classifiers respectively. Each sub-image was passed through the cascade detector in order to determine whether an eye was present within the sub-image. Images which were considered to potentially consist of an eye were passed onto subsequent stages of the cascade until a definite determination on whether the image contained an eye could be made. If a sub-image was considered not to contain an eye, the image was eliminated from the cascade. An illustration of how the cascade worked is shown in Figure 3. 

Each cascade stage is an AdaBoost classifier using Haar wavelets as features and decision stumps as weak classifiers within AdaBoost. A decision stump is a simple threshold rule computed over a single feature (a Haar wavelet). A Haar wavelet feature is a weighted sum of pixel values lying within two, three or four connected rectangles. Figure 4 shows the five possible feature shapes. Each shape has 4 trainable shape parameters: x and y offset from the origin of the window and the width and height of the rectangles. 

The features are evaluated by summing up the pixel values under the white rectangles and by subtracting the sum of the pixel values under the black rectangles (suitably weighted to normalize area). There are over 150,000 possible features which are evaluated exhaustively on the training set so that the best features are chosen at each cascade stage. A threshold is automatically determined to maximally separate positive and negative training examples, both within each weak classifier and within a single cascade stage. A decision stump, D, is evaluated on a window, w, according to the following equation:(3)D(w)= If F(w)<t then v1 else v2
where *F* (*w*) evaluates the Haar feature on the window *w* (a weighted sum of the pixel values in each rectangle), *t* is the learned threshold, and *v*1 and *v*2 are the values returned depending on if the feature is above or below the threshold. The cascade stage is evaluated as the sum of all weak classifiers in the stage as follows:(4)$$S\left(w\right)=\;D_{1}\left(w\right)+\;D_{2}\left(w\right)+\;D_{3}\left(w\right)+\ldots$$
If *S*(*w*) is greater than the stage threshold, then the window passes on to the next stage, and if not, it is rejected as outlined in Figure 3.

For example, the first cascade stage has 3 weak classifiers and eliminates approximately 60% of sub-windows. In this case, the three weak classifiers are as follows:(5)$$D_{1}\left(w\right)=\;If\;F_{1}\left(w\right)<0.081\;then-1.0\;else\;0.6\;D_{2}\left(w\right)=\;If\;F_{2}\left(w\right)<0.068\;then-0.95\;else\;0.59\;D_{3}\left(w\right)=\;If\;F_{3}\left(w\right)<0.056\;then-0.97\;else\;0.42$$

The features F_1_, F_2_ and F_3_ are shown in Figure 5. As can be seen from the figure, the features tend to find areas where there is high contrast in the target class, and this is typical of all the features used by such classifiers.

Each cascade layer was trained to achieve a true positive rate of 0.99 and a false positive rate of 0.4. Therefore, the theoretical accuracy of the entire 8-stage cascade was a true positive rate of 0.99^8^ = 0.92 with a false positive rate of 0.4^8^ = 0.00066. Example images from cascade stage 8 indicating the location of the maximum eye temperature are shown in Figure 6.

In addition to the eye detection component of the algorithm, we further developed the algorithm in order to detect the cheek region. The development of the cheek component of the algorithm required a set of training images. Training images were collected manually using a handheld infrared camera (ThermaCAM S60, FLIR Systems AB, Danderyd, Sweden) during a previous trial [1]. A total of 465 training images was collected from 43 calves, within which the cheek region was manually selected within each image by tracing a circle over the cheek muscle using ThermaCAM Researcher software (version 2.10; FLIR Systems AB, Danderyd, Sweden) (as described by Lowe et al. [1]). Based on the location of the cheek region as specified in the manually analysed images, the eye was used as a reference point in order to train the algorithm to determine the location of the cheek. The cheek region was identified by the algorithm tracing a rectangle down from the eye as shown in Figure 7. At the base of this rectangle, 3 × 3 and 9 × 9-pixel areas were automatically traced by the algorithm, from which the maximum temperatures of those areas were generated (Figure 8). The location of the cheek being determined using the eye as a reference point allowed the eye and cheek temperatures to be collected from the same images. 

The algorithm developed consists of two modes: 1) single-image mode and 2) multiple-image mode. In single-image mode, each image is treated independently and, hence, temperatures from all detected eyes or cheeks are reported; this mode is most useful when only a few images of each animal are recorded. In multiple-image mode, the algorithm assesses all images on an individual animal basis, reporting the median of the maximum eye or cheek temperatures across all images as the temperature for that animal. Multiple-image mode is most advantageous when numerous images of the same animal are being recorded. 

For both modes of image analysis, the eye component of the algorithm records the maximum temperature in degrees Celsius in two ways: 1) as the maximum temperature measured from the hottest pixel located within the eye region (algorithm: eye maximum temperature) and 2) as the maximum temperature measured from the hottest pixel within the entire image (not necessarily the eye) (algorithm: image maximum temperature). Similarly, the cheek component of the algorithm also records the maximum temperature in degrees Celsius in two ways: 1) as the maximum temperature within a 3 × 3-pixel area (algorithm: cheek 3 × 3 pixel maximum temperature) and as the maximum temperature within a 9 × 9-pixel area (algorithm: cheek 9 × 9 pixel maximum temperature).

### 2.2. Validation of the Eye and Cheek Algorithm as an Automated Method for Thermal Infrared Image Analysis

In order to validate the eye and cheek algorithm, thermal infrared images were collected and analysed automatically and compared to manual methods of analysis as described below.

#### 2.2.1. Animals

This validation component of the study was undertaken at a farm in the Waikato region of New Zealand (38°04′15.6″S 175°19′42.5″E) from July to October 2016. One hundred and twenty mix breed heifer calves (66 dairy calves (Friesian, Jersey and cross breeds) and 54 Hereford calves) (36.4 ± 4.33 kg, mean ± SD) were enrolled into the study at two days of age and were observed until 24 ± 14.4 (mean ± SD) days of age. 

#### 2.2.2. Automated Thermal Infrared Image Collection and Analysis

Throughout the course of the study, calves were fed whole milk using two automated calf feeders (RFID Calf Feeder, A&D Reid, Temuka, New Zealand (as used previously by Lowe et al. [15])). Thermal infrared images were collected automatically using an infrared camera (Thermovision A300 (accuracy: ±2.0 °C; sensitivity: <0.05 °C; resolution: 320 × 240; temperature range: −20 °C to 120 °C; spectral range: 7.5–13 µm); FLIR Systems AB, Danderyd, Sweden) integrated into each calf feeder. The left side panel of the calf feeder was modified by cutting out a square viewing hole so that the infrared camera could be placed in such a position that, as the calf fed, it could continuously collect images of the facial region during each visit (Figure 9). Thermal infrared images were captured at 60 frames per second with a resolution of 320 × 240 pixels. Calves were individually identified using an automatic electronic identification (EID) reader (G03113 EID tag reader controller R; Gallagher, Hamilton, New Zealand) and antenna system (G03121 EID tag reader antenna panel 600; Gallagher, Hamilton, New Zealand) as they entered the feeder based on the EID in their ear tags. The infrared camera was programmed to begin capturing images once the EID of the calf visiting the feeder had been detected. The infrared cameras were connected to a laptop which, through interface software, enabled the individual tag information and thermal infrared images to be collected and stored. For consistency, all images were collected at a set distance of 0.5 m and at an angle of 90° to the animal. Each infrared camera was set at an emissivity of ε = 0.98, which is accepted as a suitable emissivity for measuring an animal’s surface temperature [34] and has been used previously in cattle studies [1,3,29]. During the present study, thermal infrared images were collected during a total of 29,637 visits to the calf feeder. A subset of 350 randomly selected images were analysed using the eye and cheek detection algorithm as described above to determine the image, eye, cheek 3 × 3-pixel and cheek 9 × 9-pixel maximum temperatures.

#### 2.2.3. Manual Image Analysis

The 350 images analysed using the algorithm were also analysed manually in order to validate the algorithm. Images were analysed manually to determine the maximum image, eye and cheek (3 × 3 and 9 × 9-pixel areas) temperatures using MATLAB analysis software (R2019b; MATLAB, MathWorks Inc., Natick, MA, USA). As shown in Figure 10, the “manual: image maximum temperature” was determined by tracing the black square around the entire image. The “manual: eye maximum temperature” was determined by tracing the green square around the eye to include the eyeball and area surrounding the eyelid. The “manual: cheek 3 × 3-pixel maximum temperature” and “manual: cheek 9 × 9-pixel maximum temperatures” were determined respectively by tracing the red and blue squares over the cheek muscle using the eye as a reference point.

### 2.3. Statistical Analysis

Using Microsoft Excel (version 16.26; Microsoft Corporation, Redmond, WA, USA), data recorded automatically using the eye and cheek detection algorithm from thermal infrared images collected in the current study were regressed against the data gathered from manual analysis of the same images. This enabled the level of agreement between the two different methods of analysis to be assessed. Bias was assessed using Bland Altman analyses. In addition, a Lin’s concordance analysis was carried out to assess the level of equality between the different types of temperature measurement.

## 3. Results

Images analysed using the algorithm were highly correlated with those analysed manually for maximum image (R^2^ = 1.00, *p* < 0.001), eye (R^2^ = 0.99, *p* < 0.001), cheek 3 × 3-pixel (R^2^ = 0.85, *p* < 0.001) and cheek 9 × 9-pixel (R^2^ = 0.90, *p* < 0.001) temperatures (Figure 11). Bland Altman analysis of the differences between the algorithm and manual analysis plotted against the average showed no evidence of any change in bias across the range of values, and the average bias was not significant for maximum image (0.00 ± 0.000), eye (0.00 ± 0.001), cheek 3 × 3-pixel (0.07 ± 0.027) and cheek 9 × 9-pixel (0.08 ± 0.031) (mean difference ± standard error of the mean (SEM)) temperatures. In addition, Lin’s concordance analysis showed strong levels of equality between the two methods of analysis for maximum image (Qc = 1.00, *p* < 0.001), eye (Qc = 0.99, *p* < 0.001), cheek 3 × 3-pixel (Qc = 0.92, *p* < 0.001) and cheek 9 × 9-pixel (Qc = 0.95, *p* < 0.001) temperatures.

## 4. Discussion

The current study provides a demonstration of an automated IRT system where infrared cameras were programmed to collect images automatically as calves fed from automated calf feeders. The system used for automatically collecting thermal infrared images in the current study is similar to that used by Schaefer et al. [24], who provided the first example of an automated IRT system for the early detection of BRD in beef calves. Schaefer et al. [24] demonstrated that IRT cameras could be used to noninvasively collect thermal infrared images as calves visited a water station. Further, the current study validated an algorithm developed as an automated method of detecting and analysing the eye and cheek regions from thermal infrared images collected from young calves. 

The strongest level of agreement between the algorithm and manual method of analysis occurred when determining the maximum image temperature. This finding reflects both methods of analysis, assessing the entire image in order to determine the temperature of the single hottest pixel within the image. The maximum eye temperature also showed a strong level of agreement between both methods of analysis. This reflects the distinct appearance of the eye, making it possible for this region to be easily distinguished from the rest of the image and therefore enabling the eye region to be selected to determine the maximum eye temperature. In comparison, whilst the cheek 3 × 3 and cheek 9 × 9-pixel maximum temperatures also showed strong agreement between both methods of analysis, the level of agreement was lower than for the maximum eye temperature. In contrast to the eye region, the cheek has a less distinguishable appearance and the region of selection is instead based on using the eye as a reference point to determine the location of the cheek, which can result in some variability in the exact location selected. This level of variability consequently leads to some reduction in the level of agreement between methods when determining the maximum temperature of the cheek region. As demonstrated in the current study, the cheek 9 × 9-pixel area showed a stronger level of agreement between methods than the cheek 3 × 3-pixel area. This finding is due to the increase in pixel area, reducing the variability in the area being selected between the two methods of analysis. Validated as an automated method of analysing the eye and cheek regions from thermal infrared images, this algorithm would support the integration of IRT into automated systems for noninvasive monitoring of animal health and welfare.

Previous studies [22,23,24] investigating the use of IRT for early detection of BRD and BVD have found IRT capable of detecting the onset of the diseases based on changes in eye temperature. Schaefer et al. [23,24] found that eye temperature increased significantly in response to the onset of BRD, and these changes were found to occur several days to a week before clinical signs of disease were apparent. Similarly, Schaefer et al. [22] found a significant increase in eye temperature in response to the onset of BVD, and this increase was found to occur as early as 1-day postinfection. The integration of IRT into automated systems on-farm would enable continuous monitoring of individual animals. For the purpose of detecting diseases such as BRD and BVD, for example, this would enable a history of baseline data to be established per animal. Baseline data is essential in order for parameters to be set in such a way that deviations from what is considered “normal” for each animal can be detected. Deviations could then be used to generate alerts to notify farmers of animals displaying early signs of disease. Early disease detection would enable farmers to identify and isolate sick animals to prevent the spread of disease and would enable treatments to be administered sooner to reduce the degree of suffering [1]. Early disease detection would facilitate decision-making abilities for farmers, would reduce costs on-farm and for the industry as a whole through reduced labour, mortalities and veterinary costs and, overall, would lead to improvements in both production and animal welfare [1]. Robotic milking systems, automated feeders and automated water stations are potential systems that could support the integration of IRT. Integration of IRT into such systems would support the simultaneous collection of IRT alongside other behavioural and physiological measures. For example, Lowe et al. [1] found that milk consumption, number and duration of lying bouts, and duration of drinking visits all changed prior to the onset of NCD and suggested that they could be useful early indicators for detecting NCD. Additionally, Lowe et al. [1] manually collected and analysed thermal infrared images from a number of anatomical locations and found thermal changes of the side and shoulder of the calf to have the best potential as early indicators of NCD; therefore, these may be other anatomical areas which are worth developing further automated detection algorithms. Additionally, it may be useful to combine thermal changes, measured using IRT, with behavioural changes in feeding and drinking behaviours to provide stronger predictive composite indicators of disease. In addition to disease, the automation of IRT could also be beneficial for other applications including genetic selection and breeding and as a method for measuring stress and pain responses. Additionally, although the algorithm discussed in the current study was developed for use in young calves, with future developments, it may have further applications for use in adult cattle and other species. 

## 5. Conclusions

In conclusion, this study reports on the development and validation of an algorithm with the ability to automatically detect and analyse the eye and cheek regions from thermal infrared images collected from calves, which is a significant step towards the integration of IRT into automated systems. It is possible that further algorithms could also be developed to automatically detect and analyse other anatomical locations. With the support of algorithms, IRT could be integrated into automated systems, where, alongside other behavioural and physiological measures, IRT could be implemented as a noninvasive method of monitoring animal health and welfare. 

## Figures and Tables

**Figure 1 animals-10-00292-f001:**
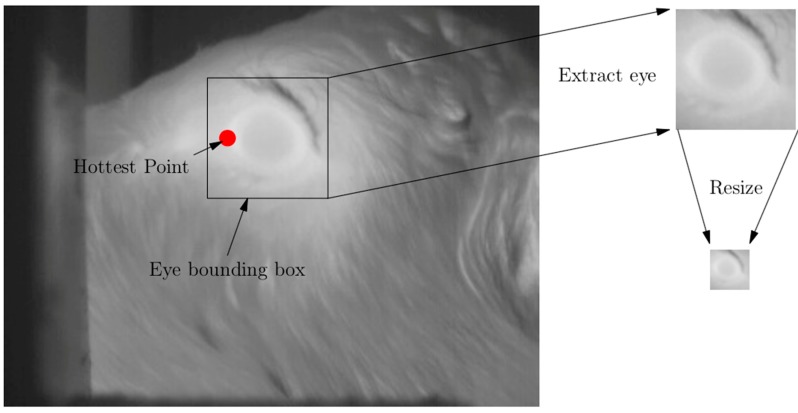
Automated thermal infrared image collected as a calf fed from the automated calf feeder showing the square the algorithm traced around the region of interest (ROI) in order to create the sub-image which was then resized for efficiency purposes to a 24 × 24-pixel image.

**Figure 2 animals-10-00292-f002:**
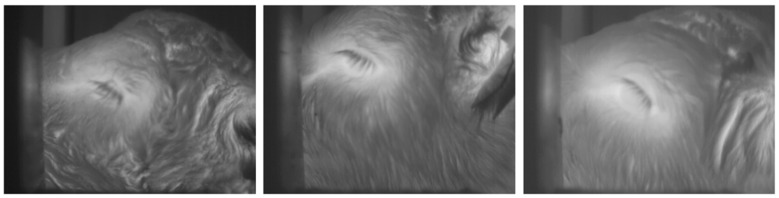
Examples of negative training images, where the eye being fully or partially closed prevented an accurate temperature measurement of the eye.

**Figure 3 animals-10-00292-f003:**
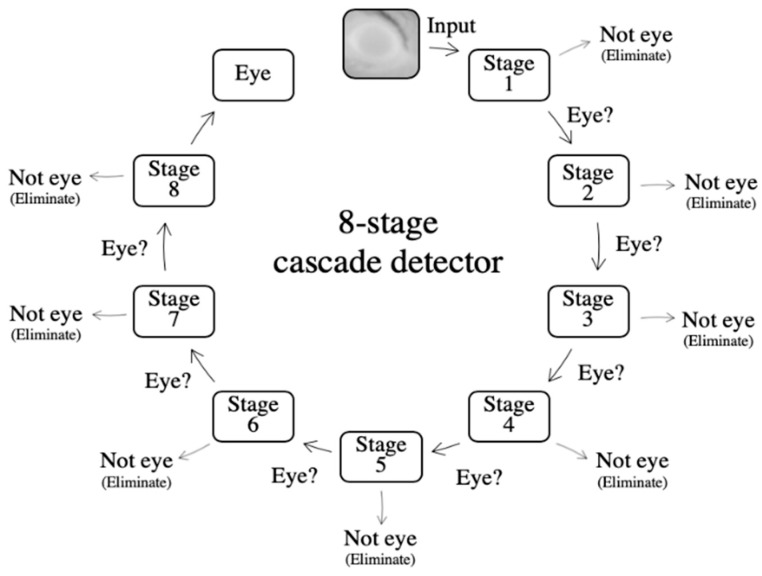
Illustration of the 8-stage cascade detector: At each stage, if the sub-image potentially contained an eye, the sub-image was passed onto the next stage of the cascade. If determined not to contain an eye, the sub-image was eliminated from the cascade.

**Figure 4 animals-10-00292-f004:**
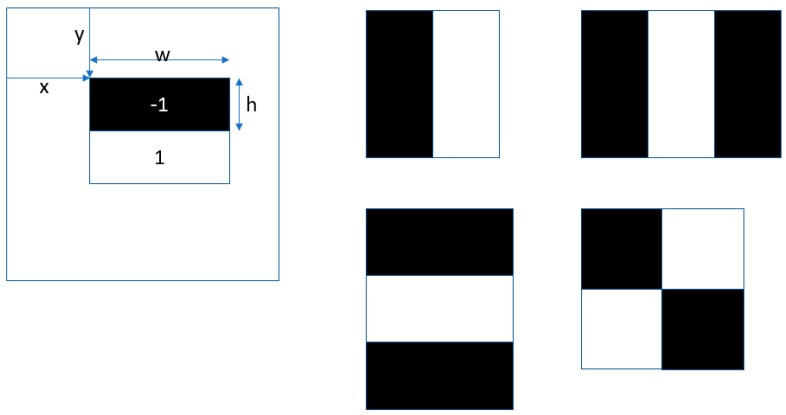
Illustration of the Haar wavelet features showing the five possible feature shapes.

**Figure 5 animals-10-00292-f005:**
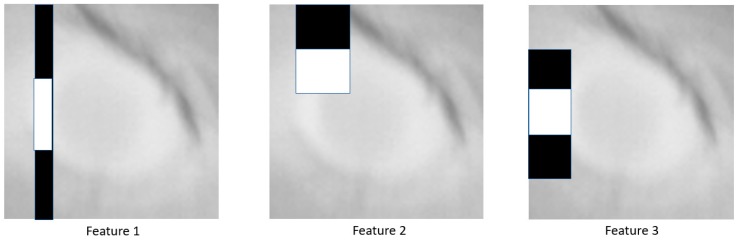
Features used in the first cascade stage.

**Figure 6 animals-10-00292-f006:**
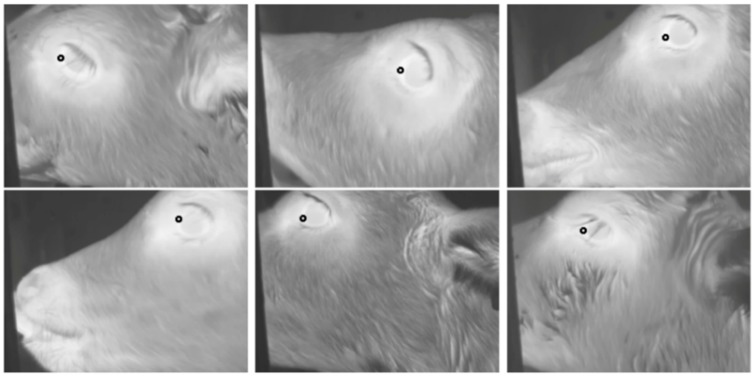
Examples of eye images from cascade stage 8: The marked positions indicate the location of the algorithm maximum eye temperature.

**Figure 7 animals-10-00292-f007:**
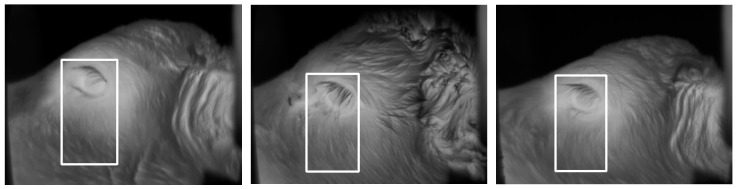
Example images which, based on the location of the eye, show the rectangle that the cheek component of the algorithm traced to determine the location of the cheek.

**Figure 8 animals-10-00292-f008:**
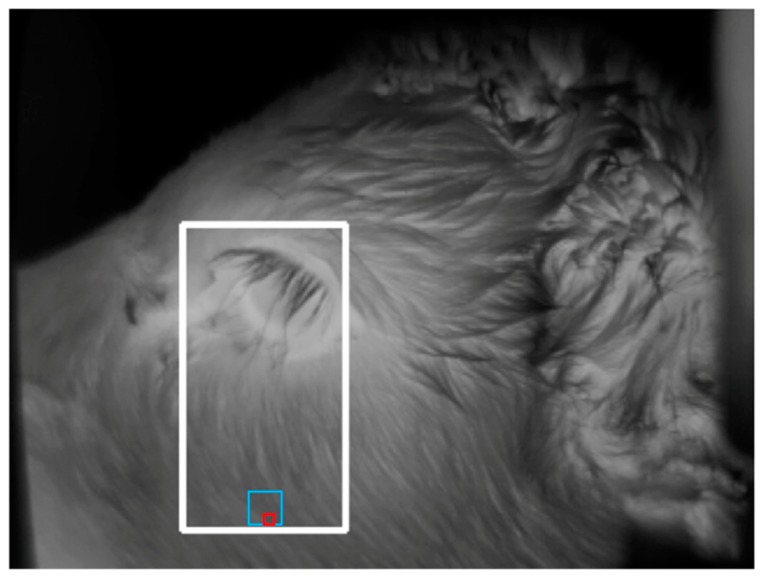
Example image showing the 3 × 3 (red square) and 9 × 9 (blue square) pixel area traced by the algorithm within the cheek ROI from which the maximum temperatures of those areas were determined.

**Figure 9 animals-10-00292-f009:**
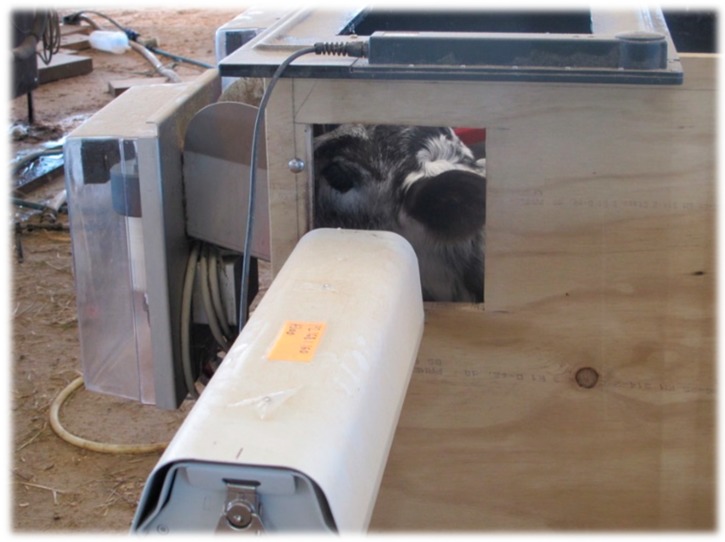
Infrared thermography (IRT) camera installed on the left side of the automated calf feeder collecting thermal infrared images of the facial region through the square viewing hole as a calf feeds from the feeder.

**Figure 10 animals-10-00292-f010:**
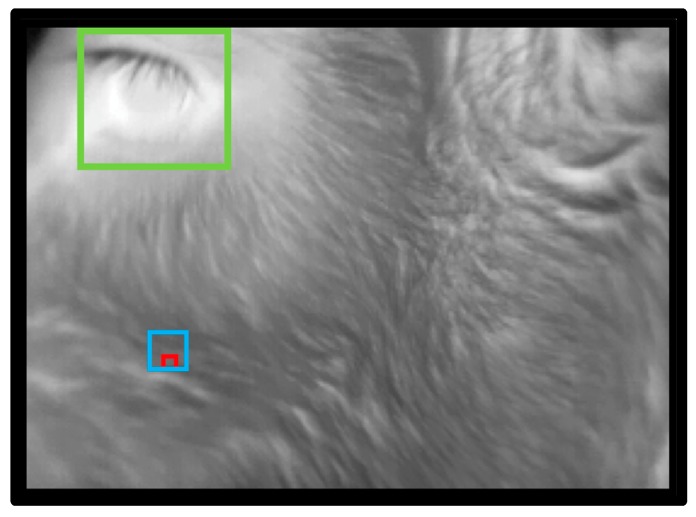
Example image showing the areas traced during the manual analysis to determine the “manual: image maximum temperature” (black square), “manual: eye maximum temperature” (green square), “manual: cheek 3 × 3-pixel maximum temperature” (red square) and “manual: cheek 9 × 9-pixel maximum temperature” (blue square).

**Figure 11 animals-10-00292-f011:**
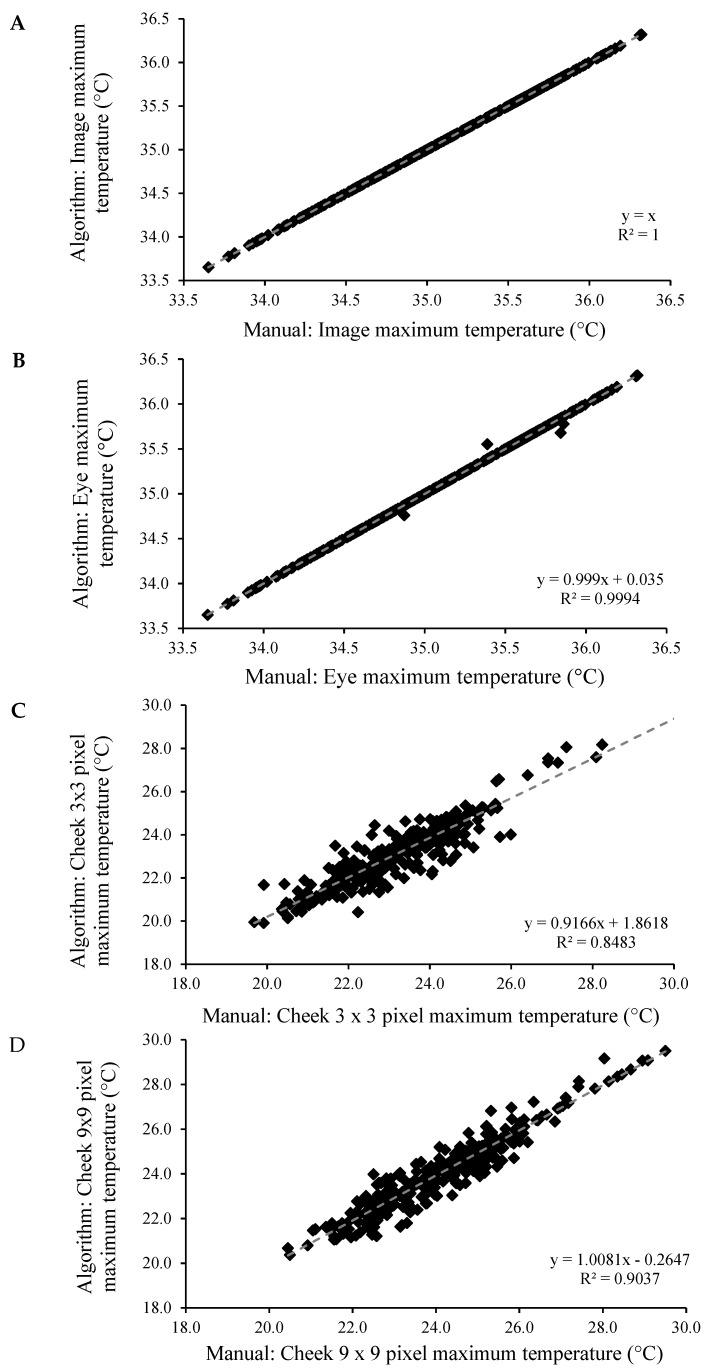
Correlations between images analysed using the algorithm and manually for the maximum (**A**) image, (**B**) eye, (**C**) cheek 3 × 3-pixel and (**D**) cheek 9 × 9-pixel temperatures.

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
