# Peer review of "Automated Collection and Analysis of Infrared Thermograms for Measuring Eye and Cheek Temperatures in Calves"

_animals, 2020, doi:10.3390/ani10020292_

Round 1
Reviewer 1 Report
Thank you for clearly addressing the reviewer comments.
Reviewer 2 Report
Thank you for incorporating my suggested changes, I think the paper reads very well.
This manuscript is a resubmission of an earlier submission. The following is a list of the peer review reports and author responses from that submission.
Round 1
Reviewer 1 Report
This study reported on the development and validation of an algorithm for automated detection and analysis of the eye and cheek regions from infrared images collected automatically from calves. However, in my opinion, there are at least two major deficiencies in the manuscript.
(1) Authors failed to describe how the thermal images were collected automatically. Equipment shown in Fig. 7 is not enough to explain the pipeline of the automatic collection of the thermal images.
(2) The algorithm to detect and analyze the eye and cheek region automatically from thermal images should be the main contribution of the manuscript. However, authors just mentioned that opencv_trainscascade was employed to detect eye region. How did you realize the 8-stage cascade? How did you optimize parameters of the opencv_trainscascade? All these important information was not provided in the manuscript.
Additionally, authors used “infrared images” many times in the manuscript. However, in my opinion, a “thermal image” is different from an “infrared image”.
Author Response
This study reported on the development and validation of an algorithm for automated detection and analysis of the eye and cheek regions from infrared images collected automatically from calves. However, in my opinion, there are at least two major deficiencies in the manuscript.
(1) Authors failed to describe how the thermal images were collected automatically. Equipment shown in Fig. 7 is not enough to explain the pipeline of the automatic collection of the thermal images.
AU: L261-282 provides details regarding how the thermal infrared images were collected automatically as calves fed from the automated calf feeder. It has been added that “The infrared cameras were connected to a laptop which through interface software enabled the individual tag information and thermal infrared images to be collected and stored.
(2) The algorithm to detect and analyze the eye and cheek region automatically from thermal images should be the main contribution of the manuscript. However, authors just mentioned that opencv_trainscascade was employed to detect eye region. How did you realize the 8-stage cascade? How did you optimize parameters of the opencv_trainscascade? All these important information was not provided in the manuscript.
AU: Additional details and supporting figures regarding the algorithm have been added to L175-207. The algorithm presented in this manuscript was developed using standard methods, the novelty lies with the application of the algorithm for the automated detection of eye and cheek temperatures in calves.
Additionally, authors used “infrared images” many times in the manuscript. However, in my opinion, a “thermal image” is different from an “infrared image”.
AU: Throughout the manuscript “infrared images” has been amended to “thermal infrared images”
Reviewer 2 Report
The topic is both an interesting and important one however I feel the authors could make the manuscript of interest to a broader audience and provide readers with a little more background by better referencing their work as suggested below:
Line 46: ref required
Line 49: additionally or 'resulting in'?
Line 50: ref required
Line 52: ref required
Line 56: ref required
Line 63: tell your reader about other species, what thermography tests for, where on the animal testing takes place and what increased/decreased temperatures indicate.
Line 66: need to explain this more fully - where the measurements were taken and give examples.
Line 67: Traditionally? Infrared imagery has not been used for long so 'traditionally' is not the correct term.
Line 74: ref required
Line 76: ref required
Line 78: never done before? How about with other species?
Line 79: ref required
Line 81: ref required
Line 89: ref required
Introduction General Comments: be clear on what you are looking for - an increase in temperature to identify disease, not stress or other use? Is this indicative of an overall body temperature increase. Refs required
Line 119: was this a pilot study?
Line 142: 'negative training' - might there be a better expression?
Line 180: be clear that this is the same image, if it is, for the eye temp and the cheek. How many images are taken/analysed to get an accurate reading?
Line 191: When might either be appropriate or 'most advantageous'?
Lines 293-299: do you require both - also see line 191
Line 309: Describe how this assists farmers and animal welfare in more detail
Line 321: Might this technology have broader implications/uses? Could it be used in other situations/with other species? Your discussion should include these broader implications/possible uses.
Line 327: Are you suggesting this is necessary?
Author Response
The topic is both an interesting and important one however I feel the authors could make the manuscript of interest to a broader audience and provide readers with a little more background by better referencing their work as suggested below:
Line 46: ref required
AU: Reference provided L49
Line 49: additionally or 'resulting in'?
AU: Because the less “hands-on” approach to farming isn’t the only factor contributing to fewer experienced stock people in the industry we have kept the use of “additionally”. L50
Line 50: ref required
AU: References provided L50-51
Line 56: ref required
AU: Reference provided L57
Line 63: tell your reader about other species, what thermography tests for, where on the animal testing takes place and what increased/decreased temperatures indicate.
AU: Further information has been added as suggested L61-67
Line 66: need to explain this more fully - where the measurements were taken and give examples.
AU: Further information has been added as suggested L73-79
Line 67: Traditionally? Infrared imagery has not been used for long so 'traditionally' is not the correct term.
AU: Amended to “Generally” L80
Line 74: ref required
AU: Reference provided L87
Line 76: ref required
AU: Reference provided L89
Line 79: ref required
AU: Reference provided L94
Line 81: ref required
AU: Reference provided L96
Line 89: ref required
AU: This sentence (L100) has not been referenced as it refers to the aim of the study.
Introduction General Comments: be clear on what you are looking for - an increase in temperature to identify disease, not stress or other use? Is this indicative of an overall body temperature increase. Refs required
AU: For the purpose of this manuscript we are simply focused on the development of an automated method of infrared image collection and analysis. Because of this we have not presented whether temperatures increase or decrease in relation to disease in the introduction as it is not the focus of this manuscript. In the discussion we do however discuss the findings of previous studies which have used IRT for disease detection.
Line 119: was this a pilot study?
AU: L132- The training images were collected as a small component of a larger study which to date have not been used in publication hence in the manuscript it is noted as being ‘unpublished data’.
Line 142: 'negative training' - might there be a better expression?
AU: L156- Our preference is to keep the use of “positive” and “negative” to distinguish between those images which contain a whole eye (positive) from those which do not contain a whole eye (negative). However, we have amended “negative training images” to “negative training examples” to be consistent with the earlier mentioned “positive training examples”.
Line 180: be clear that this is the same image, if it is, for the eye temp and the cheek. How many images are taken/analysed to get an accurate reading?
AU:It has been clarified that the eye and cheek temperatures were collected from the same images. As the location of the cheek is reliant on using the eye as a reference point the cheek temperature can only be collected from images in which the whole eye is present L226-228.
Line 191: When might either be appropriate or 'most advantageous'?
AU:The scenarios when each mode of image collection (single or multiple mode) are most appropriate or advantageous in outlined in L236-242
Lines 293-299: do you require both - also see line 191
AU: The inclusion of both the 3x3 and 9x9 pixel areas in the manuscript was to highlight how the different pixel areas impact the correlation between the automated and manually analysed images. Increasing the pixel area was found to improve the correlation and we feel it is worth including this in the manuscript to highlight the size of the pixel area as something for future research to consider for future applications. Similarly, we feel it is worthwhile including information on both the single and multiple image modes of the algorithm as for future research there may be certain situations when a particular mode is better suited to the situation.
Line 309: Describe how this assists farmers and animal welfare in more detail
AU: Further detail as to how early disease detection can assist farmers and animal welfare has been added. L364-368.
Line 321: Might this technology have broader implications/uses? Could it be used in other situations/with other species? Your discussion should include these broader implications/possible uses.
AU:The potential for this technology to be used for other applications has been added. L379-383
Line 327: Are you suggesting this is necessary?
AU: L389- We are unsure if you are referring to the support of algorithms or the collection of IRT alongside other behavioural and physiological measures. The development of algorithms will be necessary to incorporate IRT into an automated system. As discussed in the manuscript the collection of IRT alongside other behavioural and physiological measures may be beneficial for creating stronger indicators for monitoring animal health and welfare e.g. disease.
Reviewer 3 Report
58-60 awkward sentence please reword. “an animal’s surface temperature [2], where the temperatures detected and the distribution”
81-86 The benefits to measuring disease accurately on-farm are provided, but one area that is not mentioned is genetic selection or breeding. Selection for health traits is a high priority in cattle and other livestock species. One of the hindrances around selection for health in cattle is simply lack of reliable phenotypes. The system evaluated could be a source of reliable, automated and consistent health phenotypes that could be integrated into a genetic improvement program with an emphasis on improving cattle health and welfare.
95-117 the methods in this section are outside my area of expertise and should be reviewed by someone with some specific expertise in this area.
149-150 “The training process resulted in an 8-stage cascade, with each stage consisting of 3, 3, 3, 5, 4, 6, 8 and 9 weak classifiers respectively.” This algorithm requires some more description. The idea of an 8 stage cascade is straightforward enough, but exactly what is happening at each point in the cascade to determine eye/no eye is not. You mention 3, 5, 4, 6, etc “weak classifiers” but no definition of a “weak classifier” is provided. More detail is required for the reader to really understand the algorithm.
Use of maximum temperature – The cheek algorithm looks for the maximum temperature in the region. Do you have evidence that this is the best metric vs. median? My concern with maximum temperature is that it can be influenced by the texture of the surface. This is partly why the corner of the eye has the highest maximum temperature, simply because there is a natural crevice there. These crevices create what looks like a higher temperature based on infrared imaging. I understand that with the cheek images you are trying to ascertain the skin temp below the hair and where the temp is hottest is where the skin is showing more beneath the hair. So this could very well be the best. I think the paper would improve with some discussion as to why the maximum temperature was used with acknowledgement of the problem created with texture and crevices, inflating these maximums.
185-191 Multiple-image mode. Does the multiple-image mode simply look at the maximum temperature across all images from the same animal and present back that one global maximum? If this is the case, I am suspicious if this will be the most predictive of the outcome such as calf health. My reason for concern is around the phenomenon of spurious maximums that are outliers and not real values. With multiple images and picking the maximum, you are essentially “combing” for these outliers. Would a median “maximum” or median “median” not be better, to avoid the outliers?
281-285 “The strongest level of agreement between the algorithm and manual method of analysis occurred when determining the maximum image temperature.” This would imply that other metrics were also evaluated, such as the median etc. But no figures were presented for analyses other than the maximum. As pointed out above, I have some concerns with maximum values from IR images and would like to see the same algorithm performance when median or some other metrics are tested.
Author Response
58-60 awkward sentence please reword. “an animal’s surface temperature [2], where the temperatures detected and the distribution”
AU: L 59-61: Sentence has been reworded.
81-86 The benefits to measuring disease accurately on-farm are provided, but one area that is not mentioned is genetic selection or breeding. Selection for health traits is a high priority in cattle and other livestock species. One of the hindrances around selection for health in cattle is simply lack of reliable phenotypes. The system evaluated could be a source of reliable, automated and consistent health phenotypes that could be integrated into a genetic improvement program with an emphasis on improving cattle health and welfare.
AU:The potential for the system to be used for applications other than disease have been included later on in the manuscript in the discussion L379-383.
95-117 the methods in this section are outside my area of expertise and should be reviewed by someone with some specific expertise in this area.
149-150 “The training process resulted in an 8-stage cascade, with each stage consisting of 3, 3, 3, 5, 4, 6, 8 and 9 weak classifiers respectively.” This algorithm requires some more description. The idea of an 8 stage cascade is straightforward enough, but exactly what is happening at each point in the cascade to determine eye/no eye is not. You mention 3, 5, 4, 6, etc “weak classifiers” but no definition of a “weak classifier” is provided. More detail is required for the reader to really understand the algorithm.
AU:Additional information and supporting figures regarding the algorithm have been added to L175-206. The algorithm presented in this manuscript was developed using standard methods, the novelty lies with the application of the algorithm for the automated detection of eye and cheek temperatures in calves.
185-191 Multiple-image mode. Does the multiple-image mode simply look at the maximum temperature across all images from the same animal and present back that one global maximum? If this is the case, I am suspicious if this will be the most predictive of the outcome such as calf health. My reason for concern is around the phenomenon of spurious maximums that are outliers and not real values. With multiple images and picking the maximum, you are essentially “combing” for these outliers. Would a median “maximum” or median “median” not be better, to avoid the outliers?
AU:Thank you for highlighting this, multiple-image mode reports back the median-maximum across all images from the same animal. The manuscript originally stated that multiple image-mode reports the maximum temperature, this has been corrected to state that multiple-image mode reports back the “median-maximum” eye or cheek temperatures. L240
281-285 “The strongest level of agreement between the algorithm and manual method of analysis occurred when determining the maximum image temperature.” This would imply that other metrics were also evaluated, such as the median etc. But no figures were presented for analyses other than the maximum. As pointed out above, I have some concerns with maximum values from IR images and would like to see the same algorithm performance when median or some other metrics are tested.
AU: The correlation between the manual and automated methods of analysis represents that the two methods of analysis are selecting the same area. If the algorithm was set to report back the median or minimum temperature or some other diagnostic value the correlations would be expected to be the same as that obtained in the current study for maximum temperature as it is the same regions/pixel areas which are being analysed. As mentioned in the manuscript previous research has reported the maximum temperature to be the most relevant diagnostic. Therefore, for the purpose of this study we focused solely on the maximum temperature of the eye and cheek regions.